# Plasticity of the Injured Spinal Cord

**DOI:** 10.3390/cells10081886

**Published:** 2021-07-26

**Authors:** Nicolas Guérout

**Affiliations:** EA3830 GRHV, Institute for Research and Innovation in Biomedicine (IRIB), Normandie Université, UNIROUEN, 76000 Rouen, France; nicolas.guerout@univ-rouen.fr; Tel.: +33-6-22-717-217

**Keywords:** spinal cord injury, plasticity, stem cells, transplantation, repair, rehabilitation, glial scar

## Abstract

Complete spinal cord injury (SCI) leads to permanent motor, sensitive and sensory deficits. In humans, there is currently no therapy to promote recovery and the only available treatments include surgical intervention to prevent further damage and symptomatic relief of pain and infections in the acute and chronic phases, respectively. Basically, the spinal cord is classically viewed as a nonregenerative tissue with limited plasticity. Thereby the establishment of the “glial” scar which appears within the SCI is mainly described as a hermetic barrier for axon regeneration. However, recent discoveries have shed new light on the intrinsic functional plasticity and endogenous recovery potential of the spinal cord. In this review, we will address the different aspects that the spinal cord plasticity can take on. Indeed, different experimental paradigms have demonstrated that axonal regrowth can occur even after complete SCI. Moreover, recent articles have demonstrated too that the “glial” scar is in fact composed of several cellular populations and that each of them exerts specific roles after SCI. These recent discoveries underline the underestimation of the plasticity of the spinal cord at cellular and molecular levels. Finally, we will address the modulation of this endogenous spinal cord plasticity and the perspectives of future therapeutic opportunities which can be offered by modulating the injured spinal cord microenvironment.

## 1. Introduction: Spinal Cord Injury from First Descriptions to Now

The first known text describing spinal cord injury (SCI) cases is the Edwin Smith papyrus, this text was written during the sixteenth century B.C. The Edwin Smith papyrus includes the description of six SCI cases among 48 wound, injury and fracture cases. It is very interesting to note that the author of the Edwin Smith papyrus considers SCI as “untreatable” [1]. Even if the description of the cellular and molecular mechanisms, as well as the histological events which take place after SCI, are considerably more detailed nowadays, there is still no treatment which can be proposed to the injured patients. Indeed, SCI leads to a permanent loss of functions below the injury level; the aftermath can be sensitive, sensorial, autonomic and/or motor according to the severity of the injury. The deficits can be present in the lower body part or in the upper and lower body parts according to the level of the lesion. That is why the spinal cord has been seen as a nonregenerative tissue with very limited plasticity. In the present review, we will see that although the injured spinal cord presents macroscopically virtually no plasticity, that is not the case at cellular level. In fact, the spinal cord possesses an underexploited endogenous stem cell population around the central canal which can be reactivated after SCI and even that the non-lesioned spinal cord may be a neurogenic tissue. Moreover, recent discoveries show that the different cellular populations which constitute the lesion scar can be manipulated, which in turn can induce axonal regrowth and functional recovery.

## 2. Experimental Lesion Models as Proof of Concept of Spinal Cord Plasticity

In order to study SCI, several lesion paradigms and different animal species have been proposed, used and described. In fact, experimental SCI models can be classified, firstly, according to their severity, such as mild or severe, secondly, according to the type of injury: contusion or transection and, thirdly, according to anatomical considerations, such as complete or incomplete. These classifications are of primary importance since the type of injury will influence the secondary event cascades. Indeed, several studies have described that axonal regrowth can occur after SCI in the case of incomplete injury such as hemisection [2]. Maier et al., in 2008, have shown in an elegant study that after a unilateral corticospinal tract (CST) injury, constraint-induced movement therapy enhances axonal regrowth. Using anterograde tracing, this study demonstrated that CST fibers can cross the midline and grow toward the contralateral denervated gray matter, inducing functional recovery [3].

The peripheral nervous system (PNS) is well known for its intrinsic regrowth capabilities. In fact, peripheral nerve (PN) injury induces degeneration of the distal part of the PN, called Wallerian degeneration, dedifferentiation of the Schwann cells present in the distal stump and axonal regrowth from the axonal growth cone present in the proximal stump of the injured nerve [4]. Researchers took advantage of these properties and propose to use them as therapy. Peripheral nerve graft (PNG) bridging model-based studies have shown elegantly that axons can regrow in an injury context. In effect, in 2002, Gauthier et al. used, for the first time, this paradigm in a cervical hemisection model [5]. They described that a peripheral nerve can be used as a graft to bridge the two stumps of a lesioned spinal cord, the spared axons which are present above the lesion grow into the graft and restore the breathing function by the denervated hemidiaphragm [5,6]. Interestingly, Alilain et al., using the same model in 2011, have shown that restored breathing function is abolished after transection of the PNG, underlining the main importance of the axonal regrowth in breathing recovery [6].

Another experimental model based on the link between the PNS and the central nervous system is the conditioning lesion paradigm. This model was described several decades ago [7], in this model a crush lesion of a PN (conditioning lesion) before SCI induces axonal regrowth into the lesioned spinal cord. This effect can also be found after the administration of chemical demyelinating agents into the PN [8]. Even if the precise cellular and molecular mechanisms which enhance axonal regrowth are not well known, this model reflects the plasticity which can take place in the injured spinal cord.

Interestingly, recently, a new conditioning model has been described. Indeed, intermittent hypoxia preconditioning has been described as a model which enhances axonal plasticity of the injured spinal cord [9].

As we just reported before, several studies have described that axonal regrowth can occur after SCI [10]. Moreover, it has been shown that this axonal regrowth is different according to the origin and nature of the fibers. Indeed, sensitive and motor tracts do not have the same ability to regrow after SCI. It is important also to distinguish axonal regrowth which can occur after incomplete and complete SCI [2].

## 3. Intrinsic Regrowth Abilities of the Descending and Ascending Fibers after SCI

Most of the studies regarding axonal regrowth investigate the plasticity of the motor fibers and in particular those of the CST [11]. The CST controls fine motor movements and is often explored via anterograde tract tracing, using mainly an adeno-associated virus or biotinylated dextran amine injections. After incomplete SCI, it is now well described that CST fibers can regrow or sprout or at least be reorganized with intraspinal neurons after SCI to reestablish functional motor circuits [12]. After complete SCI, regrowing of the CST fibers is way more challenging. Several causes can be put forward, such as the lack of an axonal supporting microenvironment. Indeed, after complete SCI in rodents, schematically two main events will strongly inhibit axonal regrowth: in mice, the presence of a fibroblastic scar and/or in rats, the presence of cystic cavities. Moreover, recently the lack of chemoattractive, below the lesion, or chemorepellent, above the lesion, molecules has been highlighted as a major regulator of axonal regrowth after SCI [10].

It is important to note that motor movements are not only controlled by CST fibers but by many other tracts, such as rubrospinal, tectospinal or reticulospinal tracts [13]. Among them, the reticulospinal tract (RST) is very interesting, indeed these tracts control automatic rhythmic movements, including walking, via central pattern generator projections. RS neurons are constituted of several subpopulations of cells, mainly glutamatergic which exert different or even opposite roles during locomotion [14]. Few studies have investigated the plasticity of the RST. Nevertheless, it has been demonstrated that after incomplete SCI, RST can sprout below the lesion site [15].

More interestingly, in contrast to the CST, RST fibers can be spared after complete or at least severe SCI [16]. Furthermore, in this study, the authors describe that these spared RST fibers after rehabilitation can relay cortical command and induce locomotion [16].

The plasticity of the ascending tracts has been less investigated. Schematically, we can differentiate three types of ascending tracts according to their main biological functions: the lemniscal tract which carries discriminative touch; the anterolateral tract which carries heating perception and nociception and the spinocerebellar tract (which can be included in the lemniscal tracts) which carries proprioception. In animal models, the functional recovery related to these specific tracts can be very hard to investigate due to the fact that some of these sensitive tracts conduct unconscious information such as proprioception. Furthermore, the anatomical organization of these tracts complicate even more the study of their respective axonal regrowth abilities. Indeed, the nuclei of the primary neuron for all these tracts are present in the dorsal root ganglions; the lemniscal tracts project directly their axons to the brain whereas the anterolateral tracts synapse directly into the dorsal horn of the spinal cord. This major difference makes the comparison of the axonal regrowth capacities of these two major tracts difficult.

Nevertheless, several studies have investigated the abilities of the axonal regrowth of the ascending fibers. Mainly in these studies, in order to study the somatosensory tract plasticity, the authors performed dorsal column injury. In this model, lemniscal tracts show an impressive ability to sprout and regrow and more importantly, to form functional synapses [17,18]. 

## 4. Cellular Populations Which Constitute the Lesion Scar

At cellular and molecular levels, the lesioned microenvironment which is put in place after SCI has also been considered for several decades, mainly as an inhibitory barrier to axonal regrowth and functional recovery. However, after 10 to 15 years, the precise roles which are played by the lesion scar after SCI are more precisely described and tend to demonstrate that the scar exerts a dual well-balanced permissive/inhibitory effect.

Indeed, the lesion scar has been initially described as a glial scar mainly composed of reactive astrocytes expressing a high level of GFAP. However, recent studies have brought new knowledge regarding the cellular populations which compose the lesion scar. Indeed, fate mapping experiments using different and specific mouse lines, show that the lesion core is composed of fibroblasts derived from perivascular cells [19]. Moreover, it was also described that these fibroblastic cells are mixed with astrocytes derived from ependymal cells [20]. These discoveries are highly important. In fact, they illustrate the cellular heterogeneity of the spinal scar, and also demonstrate that the spinal cord possesses an intrinsic stem cell potential. In effect, Frisen’s group’s studies have shown that ependymal cells which surround the central canal are quiescent in an uninjured spinal cord but become reactive after SCI. These cells, in the injury context, proliferate, migrate into the lesion site and give rise to low expressing GFAP astrocytes [21,22]. Moreover, a small amount of the ependymal cells’ progeny differentiates into myelinating oligodendrocytes. The same team has also stressed the presence of fibroblasts in the lesion core of the spinal scar [19]. It is important to note that the reactivity of these different cellular populations and their contribution to the scar were first revealed in transection models. Nevertheless, these results have been reproduced by other teams after contusive SCI [23,24]. Recently, a study, based on single cell experiments, has described the appearance of an astroependymal cell population present only after SCI in mice, confirming the reactivity and the differentiation potential of the ependymal cells in the lesioned spinal cord [25].

Another very important population which contributes to the formation of the spinal scar is the reactive astrocyte population. These cells are located around the lesion core, highly expressed GFAP and surround the fibroblasts derived from pericytes and the astrocytes derived from ependymal cells [20]. Reactive astrocytes have been seen since the first histological description of the lesion scar as the main component of it, thereby the scar has been described as a “glial” scar. The glial scar is classically viewed as being responsible for the lack of axonal regrowth and functional recovery within SCI because reactive astrocytes secrete, among other molecules, chondroitin sulfate proteoglycan (CSPG) known to inhibit axonal regeneration. Nonetheless, recently, a very elegant study in which the authors, by attenuating or ablating scar-forming astrocytes, demonstrate that the “glial” scar prevents axonal dieback and promotes the spontaneous regrowth of the descending and ascending fibers which can be observed after SCI [26].

These results perfectly illustrate the underestimated cellular heterogeneity of the lesioned spinal cord. We can hypothesize that pericytes, astrocytes and ependymal cells are constituted of different subpopulations which will be characterized in the near future due to the increasing development of omics technologies, such as spatial proteomics or single cell RNA sequencing [25,27]. Indeed, recently, single nucleus sequencing technology identified the presence of newborn neurons in an adult uninjured spinal cord. In fact, by cell sorting EdU positive cells at different time points after EdU treatment, and by analyzing the dynamic of the transcriptional changes on these individual dividing cells, researchers revealed the presence of a discrete stem cell population which gave rise to immature and mature newborn neurons [28].

## 5. Modulation of the Spinal Cord Plasticity

Before speaking in more detail about the modulation of the spinal cord plasticity per se, it is important to note that all the previously discussed parts are linked between them. Indeed, axonal regrowth is linked to the composition of the spinal scar, and we can reasonably assume that modulating the scar will also modulate the axonal regrowth. That is why several studies had as their main aim to modulate or modify the lesioned microenvironment. At first, researchers modulate the microenvironment at molecular level in searching to reduce the expression of key inhibitory molecules. In fact, as we described previously, CSPG are known as one of the major inhibitory components of the lesion scar. That is why numerous studies investigate the effects of chondroitinase ABC (ChABC) as a treatment after SCI in order to cleave CSPG. In rodent models, this treatment has shown very promising results. In effect, administration of ChABC improves axonal regrowth and enhances functional recovery after SCI [29]. This treatment has been used alone and in co-administration with transplantation of stem cells or differentiated cells [30,31]. We can also cite another treatment which was widely investigated, and which tried to reduce the inhibitory microenvironment of the lesioned spinal cord. In fact, after SCI, the main cause of the loss of the motor, sensitive and sensory functions is the disruption of the axonal tracts, which isolate the periphery from the CNS. It also induces a massive release of axonal and myelin debris. Among this myelin debris, myelin-enriched membrane protein Nogo-A has been identified as a major axonal regrowth inhibitor, which is why Anti Nogo-A strategies (neutralization by functionally blocking antibodies, genetic deletion of Nogo-A, or blockade of Nogo-A receptors) have been proposed and investigated. After SCI, anti Nogo-A treatments induce axonal regrowth and functional recovery in rodent animals [32,33]. As we described for ChABC, anti Nogo-A therapies have also been used alone or in association with other treatments, such as ChABC [34].

Anti Nogo-A treatment has already been applied in humans after acute SCI in a feasibility study and has shown no major side effects [35].

We can also cite another molecular pathway known to inhibit axonal regrowth. In effect, the Rho pathway has been described as inducing growth cone collapse after SCI [36]. Thereby, a Rho inhibitor such as Cethrin has been tested to promote axonal regrowth and functional recovery after SCI [36]. The clinical evaluation of this pharmacological inhibitor of the Rho pathway is ongoing [37].

Thereafter, modulation of the lesioned microenvironment has been investigated at a cellular level. Indeed, knowledge acquired recently about the cellular heterogeneity of the scar allows therapies to be proposed which aim to modulate specifically the cell types present in the lesion site. This can be achieved by several approaches, such as genetic, chemical or noninvasive approaches. In fact, reducing the fibroblastic component of the scar, which is mainly an inhibitor, can induce axonal regrowth and functional recovery. Dias et al. have demonstrated it, using a specific mouse line, Glast-CreER^T2^-Rasless, which allowed them to specifically reduce the proliferation of the pericytes after SCI, that modulating the scar by reducing the PDGFRβ positive part in the lesion enhanced axonal regrowth of CST and induced sensorimotor recoveries [38].

To date, the precise role played by microglia and macrophages after SCI is still not clearly understood. Several reasons can be stated to explain it. In fact, it is very difficult to precisely discriminate macrophages and microglial cells due to their common expression of several markers. Furthermore, the inflammatory processes which take place after SCI are time dependent, meaning that the number and the profile of these cells evolves over time after SCI [39]. Moreover, microglial cells and macrophages are constituted of different subpopulations, generally described as M1 and M2, expressing, respectively, either pro- or anti-inflammatory factors. However, even if the role of microglia and macrophages after SCI is still ambiguous, it is admitted that macrophages are mainly present at the epicenter and that microglia cells constitute the border of the lesion [40]. It is also admitted that the polarization of the macrophages/microglia plays an important role in SCI, and in particular, the orientation of the microglia/macrophage’s cells to an M2 phenotype increases functional recovery and tissue repair [41].

Nevertheless, several studies using chemicals to specifically deplete microglia have brought some answers to the role played by microglia after SCI. Indeed, Lacroix’s lab has recently demonstrated the crucial role played by microglia in the contusive injury paradigm. Indeed, using a chemical approach to deplete microglia (PLX5622), they demonstrate that the lack of microglia alters the spinal scar and worsens the motor deficits [40]. To do so, they have used a colony stimulating factor 1 receptor (CSF1R) inhibitor specifically to deplete the microglia. These results have been confirmed by another study [42]. In this study, the authors also used a CSF1R inhibitor (PLX3397) to deplete microglia, they also describe that depleting microglia exacerbates functional recovery and tissue repair [42].

It is very interesting to note, that the year after, using the same chemical agent (PLX5622), another research group reached the opposite conclusion [43]. In effect, in this study the authors show that depleting microglia after contusive SCI enhances functional recovery [44]. Interestingly, in this article, the authors explore a new field of investigation at the border between regenerative and cognitive sciences. In fact, they analyze the effects of SCI on neurological functions via behavioral tests. Their results show that SCI impairs neurological functions and that depleting microglia reduces these effects [43].

These discrepancies regarding the results of these three studies can, perhaps, be explained by the differences in their methods. Indeed, in the two first studies [40,42], functional recoveries were investigated when microglia was depleted before and after SCI, whereas in the third study [43], the functional recoveries were investigated when microglia was depleted only after SCI.

Moreover, the third study [43] is the only one using male mice whereas the first study used both male and female and the second one used only female mice [40,42]. The role played by microglia on behavior, according to the sex, has been underlined recently [43]. Indeed, in this publication, the authors describe that gestational cold stress alters social behavior in adult male mice in a microglia-dependent manner [44].

These different results make it difficult to conclude about the precise role played by microglia after SCI. However, according to these different studies it can be presumed that the first week after SCI seems the most crucial time window for modulating the microglia, this period corresponding to the peak of the proliferation of these cells [40].

Noninvasive approaches such as magnetic stimulation have also been investigated recently to manipulate the cellular populations which compose the scar after SCI. In effect, in recent studies repetitive trans-spinal magnetic stimulation (rTSMS), a noninvasive therapy, has been applied to a complete transection model in mice [45,46]. In this model, rTSMS was applied 10 min per day for 14 days after SCI. The authors showed that this treatment modulates the scar by decreasing the fibrotic component and enhancing the glial component of it which induces axonal regrowth and sensorimotor recovery. 

## 6. Conclusions, Perspectives and Clinical Relevance

To conclude, the spinal cord, and in particular the injured spinal cord, possesses an underestimated and underexploited plasticity. The increasing knowledge regarding the microenvironment after SCI could in the near future bring new potential therapies. Several limitations still need to be overcome. Indeed, most of the studies which try to evaluate axonal regrowth after SCI and/or treatment are often blind about the nature, the origin and the targets of the regrowing fibers. Moreover, to date, after SCI, no treatment can induce axonal regrowth of selective fibers. Thus, several studies have reported that axonal regrowth can be correlated to allodynia or neuropathic pain [47]. In order to know and understand not only the targets of the regrowing fibers it will be important also to stress the functionality and, in particular, the relevance of these fibers during locomotion. This question could be addressed with gain and loss of function approaches using optogenetic or chemogenetic tools [48,49].

Another very important question which needs to be investigated is the precise role played by the different populations and subpopulations of cells in the establishment of the inhibitory microenvironment at the lesion site. Indeed, the main events and the overall implicated cells are known, however, the precise role played by astrocytes, ependymal cells, microglia, macrophages, pericytes, e.g., subpopulations, are not clearly known. Moreover, the lesioned spinal cord microenvironment evolves over time after SCI, it could be very important to investigate the evolution of the different cellular populations at acute and chronic phases which could be achieved by spatial mapping using single cell/single nucleus sequencing experiments [50].

An associated question is also the role of the cells implicated in the lesion scar during aging. In fact, SCI nowadays in occidental countries, concerns not only teenagers and young adults but also the elderly [51]. Few studies have inquired about the effects of SCI during aging. What is already known is that the spinal cord stem cell potential decreases over time [52]. The investigation about the other cell types is of primary importance to propose personalized treatment according to the age of the injured patient.

Finally, it is important to put into a clinical perspective all these recent discoveries and knowledge which have been mainly reported in rodents. Indeed, there are several differences between humans and rodents regarding SCI. First, in humans, SCIs are mainly due to falls or traffic accidents, meaning that injuries are mainly contusive or compressive [51]. Moreover, in humans, between half to two thirds of the SCIs are incomplete and more than half of them happen at cervical level [2]. In the preclinical model, SCIs are rarely performed at cervical level due to possible respiratory complications, most of the experimental SCIs are performed at thoracic level and mostly by transection [53]. Another difference is also the location and the relative importance of the different anatomical tracts [54]. Indeed, the descending neuronal tracts are not positioned in the same parts of the spinal cord and the CST is more developed in humans than in rodents [54]. These differences between humans and rodents are particularly important. In effect, as discussed above, the intrinsic regrowth capacities of CST are limited, however, on the other hand, the rerouting of the axonal tracts or even axonal regrowth has been described after incomplete SCI. Thereby, modulating the descending and ascending tracts after incomplete SCI in humans could lead to functional recovery. Epidural electrical stimulation (EES) has been already investigated in humans in order to initiate locomotion in injured patients [55]. This treatment has shown promising results. However, this technique is invasive and it has been reported that EES can lose its effect over time, due to the encapsulation of the electrodes by fibrous tissue [56].

The comparison of the cellular events and the establishment of the spinal scar, which takes place after SCI, should also be compared between humans and rodents. As described above, in mice the spinal scar is composed of a fibrotic component and presents no cystic cavity even several months after SCI. Whereas in rats, the first weeks after SCI the spinal scar is composed of a fibrotic component in the transection model and also in contusive SCI. However, several months after SCI the fibrotic component of the scar decreases and the spinal cord presents cystic cavities in both penetrating and contusive injuries [57]. These differences are of primary importance because they illustrate the heterogeneity of the cells which compose the scars between species. The knowledge regarding the cellular heterogeneity of the lesioned spine is not well characterized in humans. For example, the reactivity of ependymal cells after SCI in humans is still under debate. Indeed, Garcia-Ovejero et al. have demonstrated using MRI, immunohistology and gene expression profiling, that the central canal is virtually absent and is mainly constituted by perivascular pseudorosettes in humans [58]. Whereas, in contrast, Ghazale et al. have demonstrated, also using immunohistology and gene expression profiling, that the central canal region in humans expresses stem cell markers [59]. Further investigations will be needed to clarify the precise structure of the central canal and the precise role which can be played by ependymal cells after SCI in humans. These questions regarding the stem cell potential of the ependymal cells in humans might be broadened to the different cells which compose the scar after SCI. We can hope that the human atlas project will unveil even partially the cellular heterogeneity of the spinal cord in physiological and in injury contexts and will provide new insights for future research [60,61].

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
