# Peer review of "Plasticity of the Injured Spinal Cord"

_cells, 2021, doi:10.3390/cells10081886_

Round 1

Reviewer 1 Report

This review paper summarizes the research related to spinal cord plasticity and the key players in spinal cord injury. The paper is generally well written and merits publication; however, the quality of the paper can be enhanced if the following minor issues are addressed.

Minor essential revisions:

1.Abbreviations: Please write all the abbreviations in full form during their first usage. (Example; Line 229, PLX5622)

2.Experimental lesion models: (Line 41)

It would be interesting if authors can comment on the extent up to which the lesion models of SCI injury are comparable to the one in humans. Also please mention the limitations and advantages of each model developed to study SCI plasticity.

3. Role of spinal microglia in SCI: (Lines 225-231)

The authors have mentioned the role of microglial cells in SCI by citing two contradictory papers using the same chemical PLX5622. What are authors comments on these contradictory results and what conclusion will authors give regarding the role of microglia in SCI. Will microglial cell activation aid in SCI recovery or will it worsen it?

Please cite more studies that have focused on studying the protective role of microglial cell activation in SCI recovery along with their negative impacts on SCI recovery. Please draw a conclusion on this and mention what further research is essential to understand their role in SCI.

4. The questions that author missed to address in conclusion:

1.There is always a dilemma on how to conclude a review article. Since the authors have deliberately summarized huge amounts of published results, it will go a long way. It would be helpful if they can provide their own thoughts that would in turn help in finding the areas that need to be addressed. For example, what are the factors that one needs to consider what research is necessary to find therapies for SCI, what are the required criteria to overcome the key challenges like astrocyte and microglial cell activation and what are the steps required for the fast transition of stem cells research in clinical trials. Though stem cell-based therapies have been studied long enough for other diseases, why there are only a handful of clinically approved therapies? What limitations are hindering their clinical translation and in what direction does the future research need to be, to make the clinical translation possible?

Author Response

Reviewer 1:

Response: First of all, we would like to thank the reviewer for his comments and questions.

Minor essential revisions:

  1. Abbreviations: Please write all the abbreviations in full form during their first usage. (Example; Line 229, PLX5622)

R: we agree with reviewer comment, and we added an abbreviations’ list.

  1. Experimental lesion models: (Line 41)

It would be interesting if authors can comment on the extent up to which the lesion models of SCI injury are comparable to the one in humans. Also please mention the limitations and advantages of each model developed to study SCI plasticity.

R: we agree with reviewer’s comment, and we discussed this point in the last part of the review.

  1. Role of spinal microglia in SCI:(Lines 225-231)

The authors have mentioned the role of microglial cells in SCI by citing two contradictory papers using the same chemical PLX5622. What are authors comments on these contradictory results and what conclusion will authors give regarding the role of microglia in SCI. Will microglial cell activation aid in SCI recovery or will it worsen it?

Please cite more studies that have focused on studying the protective role of microglial cell activation in SCI recovery along with their negative impacts on SCI recovery. Please draw a conclusion on this and mention what further research is essential to understand their role in SCI.

R: we agree with reviewer’s comment, and we discussed this point in the review.

  1. The questions that author missed to address in conclusion: There is always a dilemma on how to conclude a review article. Since the authors have deliberately summarized huge amounts of published results, it will go a long way. It would be helpful if they can provide their own thoughts that would in turn help in finding the areas that need to be addressed. For example, what are the factors that one needs to consider what research is necessary to find therapies for SCI, what are the required criteria to overcome the key challenges like astrocyte and microglial cell activation and what are the steps required for the fast transition of stem cells research in clinical trials. Though stem cell-based therapies have been studied long enough for other diseases, why there are only a handful of clinically approved therapies? What limitations are hindering their clinical translation and in what direction does the future research need to be, to make the clinical translation possible?

R: We tried to conclude in putting into a clinical perspective these discoveries. We decided to not discuss the results obtained regarding stem cell transplantations because the precise mechanisms are not clearly understood.

Reviewer 2 Report

Nicolas Guerout presented an interesting review discussing about neuronal plasticity after spinal cord injury. It is a concise review with information that may be of interested of many researchers in the field. However, there are a few points that need to be improved.

Major

1 – The author stated that the endogenous stem cells population around the central canal can be reactivated, however, this seems to be a phenomenal only happing on animal models. The author needs to enrich the discussion on this topic, namely by mentioned the work of Daniel Garcia-Ovejero (https://doi.org/10.1093/brain/awv089 ; https://onlinelibrary.wiley.com/doi/10.1002/path.5151)

2- The author should also mention in the review the pathophysiological differences between mice, rat and human spinal cord injury, namely with regard to glial scar formation and plasticity.

3- The section of microglia/immune response on SCI neuroplasticity should be improved. There is a lot of literature on this subject that should be mentioned (microglia polarization, unbalance immune response, …) and the author almost just focused on the depletion of microglia. Moreover, the work on depressive-like behaviour and memory cited by the author is not particularly interesting for the scope of this review, there is many interesting works on microglia and SCI plasticity.

4- Authors failed to mentioned some novel therapies that are known to promote neuroplasticity after SCI, some of them are already being tested in humans (for instance EES) or shown promising results in animal models (for instance Rho-ROCK inhibition).

Minor

1 – In the beginning of the abstract and in line 30, the author mentioned that SCI leads to permanent loss of function/deficits.  Please add the word “Complete” before “spinal cord injury”. Incomplete SCI may not lead to permanent loss of function.

2- Between line 100 and 110 there is some confusion using the shortening RST, does the author mean the rubrospinal tract or the reticulospinal tract? Please clarity.

3- However it is possible to find in the literature the abbreviation C-ABC for Chondroitinase ABC, the most commonly used acronym is ChABC. I suggest changing the abbreviation.

4- It is missing a reference on line 223 in the end of the sentence “Furthermore, the inflammatory processes which take place after SCI are time and injury dependent, meaning that the number and the profile of the cells present evolves overtime and is different after penetrating or contusive SCI.”

5- The manuscript as a whole needs much more rigorous editing for spelling and grammatical errors.

Author Response

Reviewer 2:

Response: First of all, we would like to thank the reviewer for his comments and questions.

Nicolas Guerout presented an interesting review discussing about neuronal plasticity after spinal cord injury. It is a concise review with information that may be of interested of many researchers in the field. However, there are a few points that need to be improved.

Major

1 – The author stated that the endogenous stem cells population around the central canal can be reactivated, however, this seems to be a phenomenal only happing on animal models. The author needs to enrich the discussion on this topic, namely by mentioned the work of Daniel Garcia-Ovejero (https://doi.org/10.1093/brain/awv089 ; https://onlinelibrary.wiley.com/doi/10.1002/path.5151)

R: we have added this reference and we discussed this point in the last part of the review.

2- The author should also mention in the review the pathophysiological differences between mice, rat and human spinal cord injury, namely with regard to glial scar formation and plasticity.

R: we agree with reviewer’s comment, and we discussed this point in the last part of the review.

3- The section of microglia/immune response on SCI neuroplasticity should be improved. There is a lot of literature on this subject that should be mentioned (microglia polarization, unbalance immune response, …) and the author almost just focused on the depletion of microglia. Moreover, the work on depressive-like behaviour and memory cited by the author is not particularly interesting for the scope of this review, there is many interesting works on microglia and SCI plasticity.

R: we agree with reviewer’s comment, and we discussed this point in the review.

4- Authors failed to mention some novel therapies that are known to promote neuroplasticity after SCI, some of them are already being tested in humans (for instance EES) or shown promising results in animal models (for instance Rho-ROCK inhibition).

R: we agree with reviewer’s comment, and we discussed this point in the review.

Minor

1 – In the beginning of the abstract and in line 30, the author mentioned that SCI leads to permanent loss of function/deficits.  Please add the word “Complete” before “spinal cord injury”. Incomplete SCI may not lead to permanent loss of function.

R: we have added “complete” to these sentences.

2- Between line 100 and 110 there is some confusion using the shortening RST, does the author mean the rubrospinal tract or the reticulospinal tract? Please clarity.

R: we modified these sentences, and we added an abbreviations’ list.

3- However it is possible to find in the literature the abbreviation C-ABC for Chondroitinase ABC, the most commonly used acronym is ChABC. I suggest changing the abbreviation.

R: We agree with reviewer’s comment and we have changed this abbreviation.

4- It is missing a reference on line 223 in the end of the sentence “Furthermore, the inflammatory processes which take place after SCI are time and injury dependent, meaning that the number and the profile of the cells present evolves overtime and is different after penetrating or contusive SCI.”

R: we modified these sentences and we added a reference.

5- The manuscript as a whole needs much more rigorous editing for spelling and grammatical errors.

R: The manuscript has been edited.

Round 2

Reviewer 2 Report

The authors adressed the majority of my concerns. There is just one issue to revise. The author must not use unpublished data to support his claims.

Author Response

Reviewer 2:

Response: First of all, we would like to thank the reviewer for his comments and questions.

The authors adressed the majority of my concerns. There is just one issue to revise. The author must not use unpublished data to support his claims.

R: we agree with reviewer’s comment, and we removed these claims.